# Measures of Information Reflect Memorization Patterns

**Rachit Bansal**[♃]
Delhi Technological University
racbansa@gmail.com

**Danish Pruthi**[♄]
Amazon Web Services
danish@hey.com

**Yonatan Belinkov**[♁]
Technion – Israel Institute of Technology
belinkov@technion.ac.il

## Abstract

Neural networks are known to exploit spurious artifacts (or shortcuts) that co-occur with a target label, exhibiting *heuristic memorization*. On the other hand, networks have been shown to memorize training examples, resulting in *example-level memorization*. These kinds of memorization impede generalization of networks beyond their training distributions. Detecting such memorization could be challenging, often requiring researchers to curate tailored test sets. In this work, we hypothesize—and subsequently show—that the diversity in the activation patterns of different neurons is reflective of model generalization and memorization. We quantify the diversity in the neural activations through information-theoretic measures and find support for our hypothesis in experiments spanning several natural language and vision tasks. Importantly, we discover that information organization points to the two forms of memorization, even for neural activations computed on unlabeled in-distribution examples. Lastly, we demonstrate the utility of our findings for the problem of model selection. The associated code and other resources for this work are available at https://information-measures.cs.technion.ac.il.

## 1 Introduction

Current day deep learning networks are limited in their ability to generalize across different domains and settings. Prior studies found that these networks rely on spurious artifacts that are correlated with a target label (Schölkopf et al., 2012; Lapuschkin et al., 2019; Geirhos et al., 2019, 2020, inter alia). We refer to learning of such artifacts (also known as heuristics or shortcuts) as *heuristic memorization*. Further, neural networks can also memorize individual training examples and their labels; for instance, when a subset of the examples are incorrectly labeled (Zhang et al., 2017; Arpit et al., 2017; Tänzer et al., 2021). We refer to this behavior as *example-level memorization*. A large body of past work has established that these facets of memorization pose a threat to generalization, especially in out-of-distribution (OOD) scenarios where the memorized input features and corresponding target mappings do not hold (Ben-David et al., 2010; Wang et al., 2021b; Hendrycks et al., 2021a; Shen et al., 2021). To simulate such OOD distributions, however, researchers are required to laboriously collect specialized and labeled datasets to measure the extent of suspected fallacies in models. While these sets make it possible to assess model behavior over a chosen set of features, the larger remaining features remain

---

[♃]Work done during a visit at the Technion, Israel. The author is now at Google Research India.

[♄]Work done while at Carnegie Mellon University, prior to joining Amazon.

[♁]Supported by the Viterbi Fellowship in the Center for Computer Engineering at the Technion.

36th Conference on Neural Information Processing Systems (NeurIPS 2022).

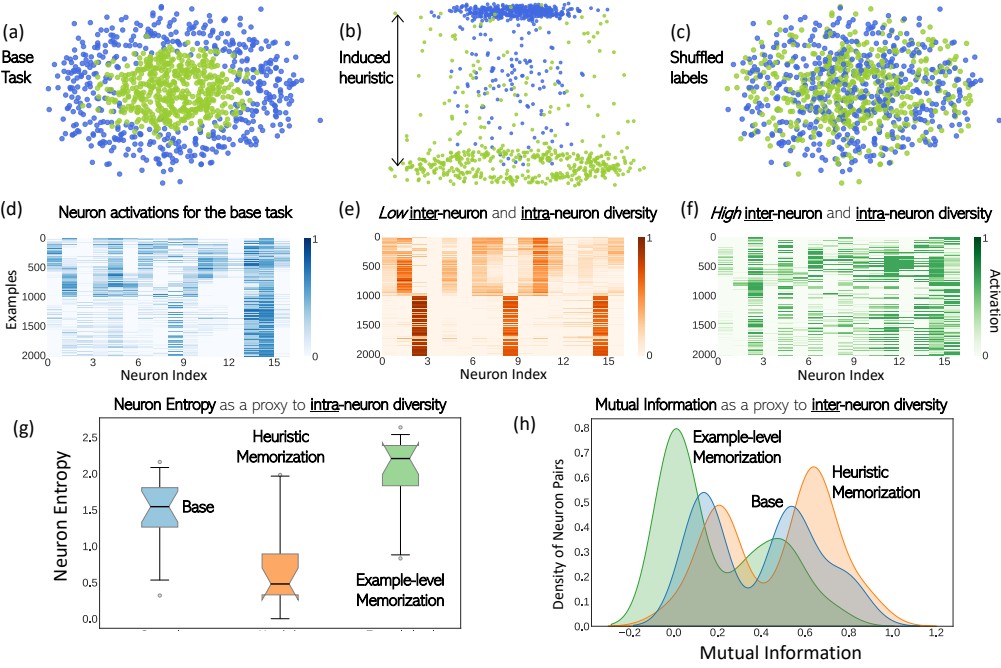

Figure 1: **(a)** A toy setup of separating concentric circles; **(b)** An additional feature spuriously simplifies the task, inciting *heuristic memorization*; **(c)** Shuffled target labels induce *example-level memorization*; **(d)** Neuron activations for a two-layered feed-forward network trained for the base task in (a); **(e)** Activation patterns for the network reflect low *intra-neuron* and *inter-neuron* diversity when trained on (b); **(f)** High *intra-neuron* and *inter-neuron* diversity is seen when the network is trained on (c); **(g)** *Entropy* acts as a proxy to intra-neuron diversity; **(h)** *Mutual Information* acts as a proxy to inter-neuron diversity. Distinguishable patterns for the three networks are seen in (g) and (h).

hard to identify and study. Moreover, these sets are truly extrinsic in nature, necessitating the use of performance measures, which in turn lack interpretability and are not indicative of internal workings that manifest certain model behaviors. These considerations motivate evaluation strategies that are intrinsic to a network and indicate model generalization while not posing practical bottlenecks in terms of specialized labeled sets. Here, we study information organization as one such potential strategy.

In this work, we posit that organization of information across internal activations of a network could be indicative of memorization. Consider a sample task of separating concentric circles, illustrated in Figure 1a. A two-layered feed-forward network can learn the circular decision boundary for this task. However, if the nature of this learning task is changed, the network may resort to memorization. When a spurious feature is introduced in this dataset such that its value $(+/-)$ correlates to the label (0/1) (Figure 1b), the network *memorizes* the feature-to-label mapping, reflected in a uniform activation pattern across neurons (Figure 1e). In contrast, when labels for the original set are shuffled (Figure 1c), the same network memorizes individual examples during training and shows a high amount of diversity in its activation patterns (Figure 1f). This example demonstrates how memorizing behavior is observed through diversity in neuron activations.

We formalize the notion of *diversity* across neuron activations through two measures: (i) *intra-neuron* diversity: the variation of activations for a neuron across a set of examples, and (ii) *inter-neuron* diversity: the dissimilarity between pairwise neuron activations on the same set of examples. We hypothesize that the nature of these quantities for two networks could point to underlying differences in their generalizing behavior. In order to quantify intra-neuron and inter-neuron diversity, we adopt the information-theoretic measures of *entropy* and *mutual information* (MI), respectively.

Throughout this work, we investigate if diversity across neural activations (§2) reflects model generalizability. We compare networks with varying levels of heuristic (§3) or example-level (§4) memorization across a variety of settings: synthetic setups based on the IMDb (Maas et al., 2011)

and `MNIST` (Lecun et al., 1998) datasets for both memorization types, as well as naturally occurring scenarios of gender bias on `Bias-in-Bios` (De-Arteaga et al., 2019) and OOD image classification on `NICO` (Zhang et al., 2022). We find that the information measures consistently capture differences among networks with varying degrees of memorization: Low entropy and high MI are characteristic of networks that show heuristic memorization, while high entropy and low MI are indicative of example-level memorization. Lastly, we evaluate these measures from the viewpoint of model selection and note strong correlations to rankings from domain-specific evaluation metrics (§5).

## 2 Methods

As per the data processing inequality (Beaudry & Renner, 2012), a part of the neural network (referred to as the *encoder*) compresses the most relevant information of a given input $X$, into a representation $H$. This compressed information is processed by a *classification head* (or, a *decoder*) to produce an output $Y$ corresponding to the given input. We hypothesize that the organization of information across neurons of the encoder is indicative of model generalization. We study two complementary properties that capture this information organization for a given network:

(i) **Intra-neuron diversity**: How do activations of a given neuron vary across different input examples. We measure the *entropy* of neural activations (across examples) as a proxy.

(ii) **Inter-neuron diversity**: How unique is the activation of a neuron compared to other neurons. We quantify this via *mutual information* between activations of pairwise neurons.

Below, we discuss the information measures formally.

### 2.1 Information Measures

For any given encoder (consisting of $N$ neurons) that maps the input to a dense hidden representation, we denote the activation of the $i^{\text{th}}$ neuron as a random variable, $A_i \in \{a_i^1, \ldots, a_i^S\}$, where each measurement is an activation over an example from a set of size $S$. The probability over this continuous activation space is computed by binning it into discrete ranges (Darbellay & Vajda, 1999), and we denote each discretized activation value as $\hat{a}$. Importantly, the set of examples on which the activations are computed come from a distribution that is similar to the underlying training set itself.

**Entropy** We measure the Shannon entropy for each neuron in the concerned network, as a proxy of intra-neuron diversity. Following the definition of Shannon entropy, this is given as:

$$H(A_i) = \underset{\hat{a}_i^s \in A_i}{\mathbb{E}} [h(\hat{a}_i^s)] = \sum_{j=1}^{N_{\text{bins}}} p(\hat{a}_i^j) \log(\frac{1}{p(\hat{a}_i^j)}) \tag{1}$$

**Mutual Information** We compute the mutual information (MI) between underlying neurons as a proxy to inter-neuron diversity. Specifically, we compute the MI between all neuron pairs in the network.[1] Thus, the set of MI values $I(A_i)$ for a particular neuron $A_i$, is given as:

$$I(A_i) = \{I(A_i; A_1), \ldots, I(A_i; A_N)\} \tag{2}$$

where, $I(X; Y)$ depicts the MI between variables $X$ and $Y$. Unless stated otherwise, this $I(A_i)$ is computed $\forall i \in \{1, \ldots, N\}$, resulting into a square matrix of size $(N \times N)$.

This process of computing the information measures for a network on a given set of examples is summarized in Algorithm 1. Further details on the computation are given in appendix A.

### 2.2 Toy Setup: Concentric Circles

Here, we briefly discuss the information-theoretic metrics for the example of concentric circles from the introduction (Figure 1). To recap, we consider a setup to compare networks showing the two forms of memorization and observe discernible differences in their activation patterns: heuristic

---

[1] In principle, we would compute MI across neuron sets; we approximate this through individual neuron pairs.

**Algorithm 1** Computation of information measures. Algorithmic procedures ENTROPY and MI are specified by algorithms 2 and 3 in appendix A.

```
 1:  A_1, ..., A_N ← {f(x_i)}_{i=1}^S                    ▷ Computing activations for all neurons
 2:  H ← {}; I ← {}                                      ▷ Initiating computations for Entropy and MI
 3:  for i ∈ {1, ..., N} do                              ▷ Iterating over the set of neurons
 4:      I_i ← {}                                        ▷ Initiating MI for a particular neuron
 5:      H_i ← ENTROPY(A_i)                              ▷ Following Equation 1 and Algorithm 2
 6:      for j ∈ {1, ..., N} do                          ▷ Inner loop over the set of neurons
 7:          I_i ← I_i ⊕ MI(A_i, A_j)                    ▷ Following Equation 3 and Algorithm 3
 8:      end for
 9:      H ← H ⊕ H_i
10:      I ← I ⊕ I_i                                     ▷ Following Equation 2
11:  end for
```

memorization corresponds to low intra-neuron and inter-neuron diversity, while example-level memorization corresponds to high diversity (Figures 1e and 1f). We expect that this difference in diversity would be captured through the above defined information measures.

Figure 1g presents the distribution of entropy values for each of the three networks with varying generalization behaviors. Throughout this work, we visualize this distribution of entropy using similar box-plots, where a black marker within the boxes depicts the median of the distribution and a notch neighboring this marker depicts the $95\%$ confidence interval around the median. We observe that entropy for the network exhibiting heuristic memorization is distributed around a lower point than the others, whereas entropy for the network with example-level memorization is higher.

Furthermore, Figure 1h shows the distribution of MI for the three networks. To interpret the distribution of MI (an $N \times N$ square matrix), we fit a Gaussian mixture model over all values and visualize it through a density plot, where the density (*y-axis*) at each point corresponds to the number of neurons pairs that exhibit that MI value (*x-axis*). Larger peaks in these density plots suggest a large number of neurons pairs are concentrated in that region. Interestingly, we see such peaks for the three networks at distinct values of MI. For the network showing example-level memorization (high inter-neuron diversity), most of the neuron pairs show low values of MI. In contrast, heuristic memorization (low inter-neuron diversity) has high neuron pair density for higher MI values.[2]

Based on these findings, we formulate two hypotheses, summarized in Table 1:

**H1** Networks exhibiting heuristic memorization would show low inter- and intra-neuron diversity, reflected through low entropy and high MI values.

**H2** Networks exhibiting example-level memorization would show high inter- and intra-neuron diversity, reflected through high entropy and low MI values.

Table 1: Summarizing our hypotheses.

| Memorization | Diversity | |
|---|---|---|
| | Intra-neuron ($\propto$ Entropy) | Inter-neuron ($\propto$ MI$^{-1}$) |
| Heuristic | ↓ | ↓ |
| Example-level | ↑ | ↑ |

## 3 Heuristic Memorization

Here, we study different networks with varying degrees of heuristic memorization, and examine if the information measures—aimed to capture neuron diversity—indicate the extent of memorization.

### 3.1 Semi-synthetic Setups

We synthetically introduce spurious artifacts in the training examples such that they co-occur with target labels. Networks trained on such a set are prone to memorizing these artifacts. The same correlations with an artifact do not hold in the validation sets. To obtain a set of networks with varying

---

[2]This difference in neuron activation patterns for the two memorizing sets could be caused by several factors, including functional complexity (Lee et al., 2020): Functions that encode individual data points (as in example-level memorization) need to be much more complex than functions that learn shortcuts (heuristic memorization). We make a comparison with standard complexity measures in appendix C.4 and observe that our information measures correlate more strongly with generalization performance—especially for heuristic memorization.

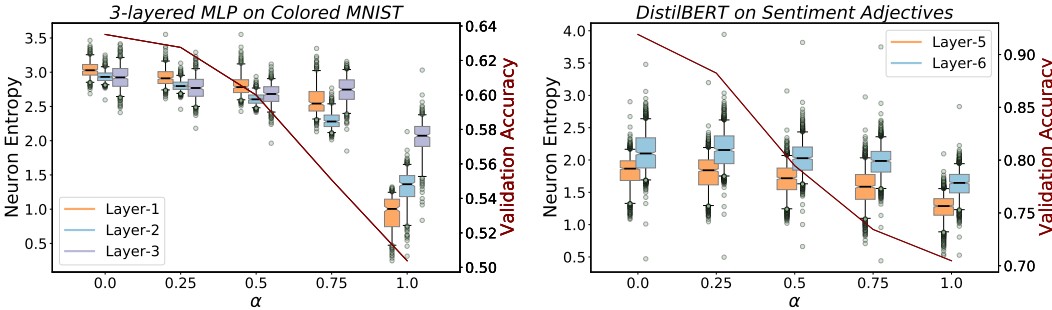

Figure 2: The relation between entropy of neural activations and heuristic memorization. For both the setups, networks trained on higher $\alpha$ show higher heuristic memorization (as depicted by the dipping model accuracy line), accompanied with lower entropy values.

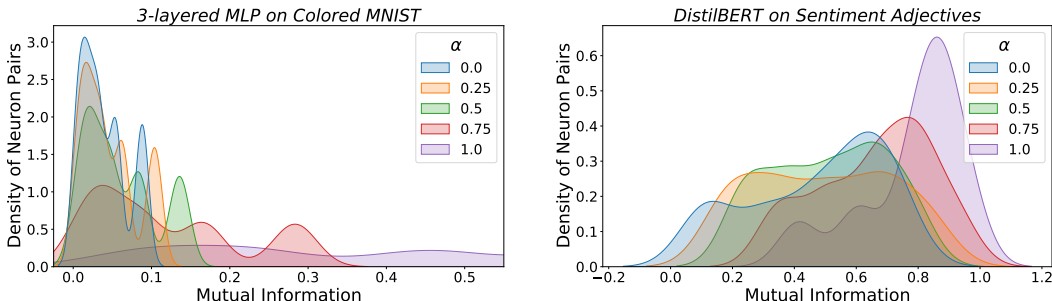

Figure 3: Distribution of mutual information (MI) of pairs of neurons for networks with varying heuristic memorization. For both settings, networks trained on training sets with larger amounts of spurious correlations ($\uparrow \alpha$) exhibit higher mutual information across their neuron pairs.

degrees of this heuristic memorization, we consider a parameter $\alpha$ that controls the fraction of the training examples for which the spurious correlation holds true. We consider the following setups:

**Colored MNIST**   In this setting, the MNIST dataset (Lecun et al., 1998) is configured such that a network trained on this set simply learns to identify the color of images and not the digits themselves (Arjovsky et al., 2019). Particularly, digits 0–4 are grouped as one label while 5–9 as the other, and images for these labels are colored green and red, respectively. For this setup, we train multi-layer perceptron (MLP) networks for varying values of $\alpha$, which corresponds to the fraction of training instances that abide to the color-to-label correlation. The considered values of $\alpha$ and other details for this setup are given in appendix B.1.

**Sentiment Adjectives**   In this setup, we sub-sample examples from the IMDb dataset (Maas et al., 2011) that contain at least one adjective from a list of positive and negative adjectives. Then, examples that contain any of the positive adjectives ("good", "great", etc.) are marked with the positive label, whereas ones that contain any negative adjectives ("bad", "awful", etc.) are labeled as negative. We exclude examples that contain adjectives from both lists. The motivation to use this setup is to introduce heuristics in the form of adjectives in the training set. We fine-tune DistilBERT-base models (Sanh et al., 2019) on this task for different values of $\alpha$ (fraction of examples that obey the heuristic). The full set of adjectives considered and further details are outlined in appendix B.2.

**Results:**   Through these experiments, we first note that **low entropy across neural activations indicates heuristic memorization in networks**. This is evident from Figure 2, where we see that (1) as we increase $\alpha$ the validation performance decreases, indicating heuristic memorization (see the solid line in the plots); and (2) with an increase in this heuristic memorization, we see lower entropy across neural activations. We show the entropy values of neural activations for the 3 layers of an MLP

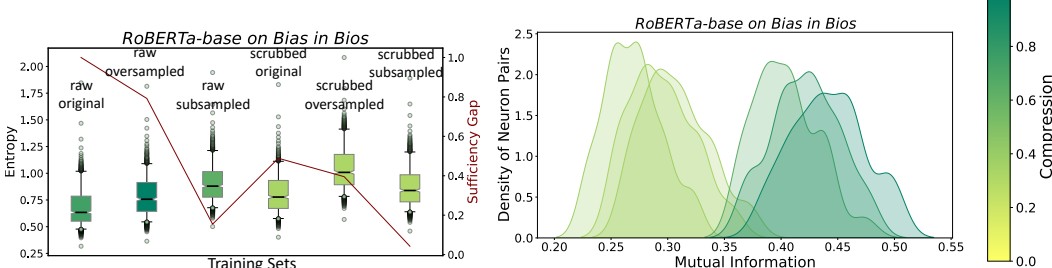

Figure 4: Distributions of entropy and MI across final layer activations of RoBERTa-base differentiate networks fine-tuned on original and de-biasing sets for `Bias-in-Bios`. Color of boxes and Gaussian plots corresponds to *extractability* of gender information in model representations as estimated through MDL probing (Voita & Titov, 2020)—lighter colors indicate lower extractability (less bias).

trained on `Colored MNIST` (left sub-plot) and for the last two layers of DistilBERT on `Sentiment Adjectives` (right sub-plot).[3] In both these two scenarios, we see a consistent drop in the entropy with increasing values of $\alpha$, with a particularly sharp decline when $\alpha = 1.0$.

Furthermore, we observe that **networks with higher heuristic memorization exhibit higher mutual information** across pairs of neurons. In Figure 3, networks with higher memorization ($\uparrow \alpha$), have larger density of neurons in the high mutual information region. While this trend is consistent across the two settings, we see some qualitative differences: The memorizing ($\alpha = 1.0$) MLP network on `Colored MNIST` (left) has a uniform distribution across the entire scale of MI values, while DistilBERT on `Sentiment Adjectives` (right) largely has a high-density peak for an MI of $\sim 0.9$.

## 3.2 Natural Setups

Next, we investigate setups where spurious correlations are not synthetically induced, but occur naturally in the datasets. Below, we describe two such scenarios:

**Occupation Prediction** We first study the task of predicting occupations from biographies on the `Bias-in-Bios` dataset (De-Arteaga et al., 2019). Given the skewed distribution of genders across occupations, models pick up cues that reveal the biographee's gender. For instance, most biographies corresponding to the "professor" occupation are of males. Models trained on this dataset can learn such spurious associations. To evaluate how much the trained networks encode gender, we measure *compression* values by training a gender classifier on the internal representations of the network and computing its minimum description length (MDL). These compression values act as a proxy to the ease of extracting gender information from representations (Voita & Titov, 2020; Orgad et al., 2022).

We consider a variety of training sets by *sub-sampling* and *over-sampling* examples for each profession in the dataset: This is done to balance the number of examples across each gender. We do this for both the original inputs in the dataset (*raw*) and *scrubbed* examples, wherein gender-specific information (such as pronouns) is removed (similar to setups in De-Arteaga et al. (2019)). We perform our analysis on RoBERTa-base (Liu et al., 2019) fine-tuned for these training sets.[4]

**Results:** In Figure 4, we observe the distribution of the two information measures for the last layer of networks trained on the different training sets.[5] This variation is shown in conjunction with compression values across the network using the MDL probe. Following our initial hypothesis (Table 1; **H1**), we expect that networks with higher representation of bias will have lower entropy. Indeed, in Figure 4 (left), the network trained on the original training set (i.e., `raw original`) shows the lowest entropy. This finding is in line with our hypothesis, since the other networks are trained on either gender-balanced or scrubbed sets. However, we do not observe consistent trends among

---

[3]Considerable changes in entropy values are not seen for initial DistilBERT layers, suggesting that spurious correlations are largely captured by later layers. Detailed results covering other layers are given in appendix C.1.

[4]We use trained checkpoints released by Orgad et al. (2022). More details are given in appendix B.3.

[5]The difference in compression values across training sets is more prominent in higher layers, yet the correlation between compression and MI remains high throughout the network. We discuss this in appendix C.3.

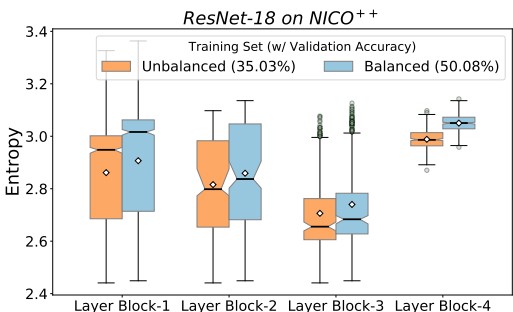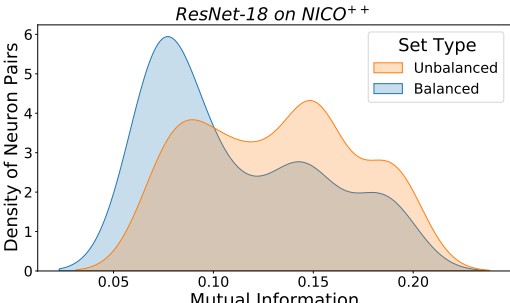

Figure 5: Entropy and MI for ResNet-18 on the `NICO`$^{++}$ dataset. The two training sets—`balanced` and `unbalanced`—result into models that vary in their generalization to contextual features beyond on what they were trained on. This distinction is reflected in the information measurements.

networks trained on these de-biasing sets. On the other hand, we do see clear patterns in MI that distinguish networks in line with their compression values (Figure 4, right). As we go from lower to higher values of MI (left to right), the density plots get darker, corresponding to higher compression values (higher bias). A prominent distinction is seen between the `raw` and `scrubbed` sets, which are separated on two sides of the plot.

**Image Classification with Contexts**   Next, we consider a scenario from computer vision, where the task is to identify the presented object in a particular context. We use a subset of the `NICO`$^{++}$ dataset (Zhang et al., 2022), which consists of images of animals in a variety of contexts. For each animal class, there exist two types of contexts: *individual*, those that are specific to only that animal and are not present for all classes (such as a *roaring* bear), and *common*, contexts that exist across all classes (such as images taken in *dark*).

For our analysis, we design two training sets—`unbalanced` and `balanced`—varying in the distribution of common contexts across examples. Each animal in the `unbalanced` set occurs in a particular common context that is chosen for that animal. In contrast, the `balanced` set contains images from all common contexts, for each animal. Thus, a network trained on the `unbalanced` set is likely to pick the context-to-animal mapping (i.e., a case of heuristic memorization).

**Results:** We train ResNet-18 (He et al., 2015) networks for the two sets and evaluate them on the common `NICO`$^{++}$ evaluation set, balanced across all common contexts. We consider the hidden representation from each of the 4 blocks of layers in the network to compute the information measures reported in Figure 5. From the left sub-figure, we observe that the entropy for networks trained on the `balanced` set is consistently greater than the `unbalanced` set across all layer blocks. Furthermore, we observe that distribution of MI (right) across pairwise neurons also reflects the difference between the networks, corroborating our hypothesis. Neuron pairs for the network that memorizes the correlation with image contexts (`unbalanced`) are more densely concentrated at higher MI values.

## 4   Example-level Memorization

We now examine how the distribution of information measures across networks change when they memorize individual examples. Following our original hypotheses (Table 1; **H2**), we expect such networks to display high intra-neuron and inter-neuron diversity, and thus high entropy and low MI.

We perform the analysis for example-level memorization on the standard datasets of `MNIST` (Lecun et al., 1998) and `IMDb` (Maas et al., 2011) on a 3-layered MLP and DistilBERT-base, respectively. In order to study how the diversity of neurons changes with increasing example-level memorization, we induce varying levels of label noise by randomly shuffling a fraction of training examples' target labels (denoted by a parameter $\beta$). We then analyze these trained networks on the original validation set.

**Results:**   First, we note that model performance on the validation set decreases with increased label shuffling, validating an increase in example-level memorization (Figure 6). Interestingly, this dip in validation accuracy is accompanied with a consistent rise in entropy across the neurons. For MLP networks trained on `MNIST` (left), we see a distinct rise in entropy even with a small amount of label

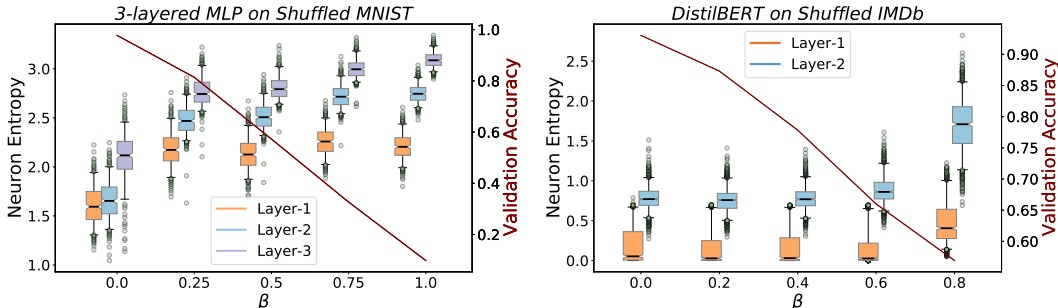

Figure 6: Entropy across neuron activations increases with greater example-level memorization ($\uparrow \beta$).

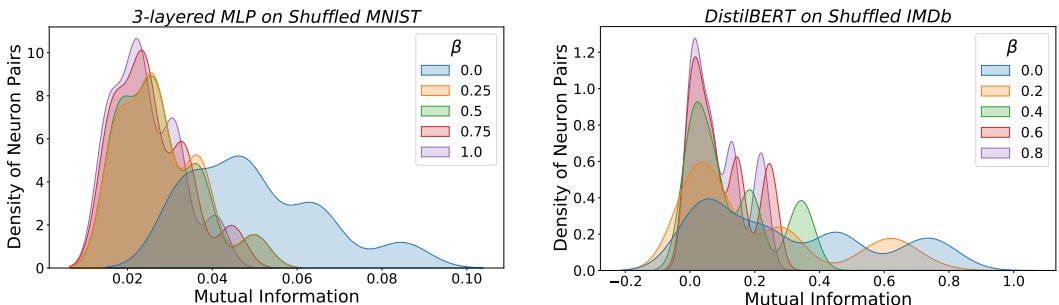

Figure 7: Networks that show higher example-level memorization ($\uparrow \beta$) have high density of neuron pairs for lower MI values. Here, MI is computed across the first layer for both the networks.

shuffling ($\beta = 0.25$), followed by a steady increase (layers 2 and 3) or no change (layer 1) in entropy. A dissimilar trend is seen for DistilBERT fine-tuned on `IMDb` (right): a distinct rise for high values of $\beta$ and a consistent value for low or no label shuffling. While our hypothesis holds true in both settings, we speculate the difference between them is due to the pre-trained initialization of DistilBERT, which has been shown to act as an implicit regularization during fine-tuning (Tu et al., 2020). That is, here, DistilBERT might be learning task-relevant information despite some amount of label noise (note that this is not evident through validation performance alone).

Our hypothesis for the relation between example-level memorization and MI is supported by Figure 7. In both settings, networks trained on higher $\beta$ values consist of neuron pairs that show low values of MI (left side of the plots). In line with the previous observations, we find that MLPs trained on some amount of label noise (any $\beta > 0.00$) on MNIST (left sub-plot) have a higher density of neuron pairs concentrated at low values of MI. Meanwhile, for DistilBERT on `IMDb` (right sub-plot), we observe that neuron pair density gradually shifts towards lower values of MI with increasing $\beta$.[6]

## 5 Model Selection

In the previous sections, we have seen that studying information organization through the presented measures allow us to qualitatively distinguish networks with different generalizing behaviors. A natural application of our findings is the problem of model selection: given a list of models, rank them based on their generalizability. To demonstrate the utility of our insights, we compare the correlations between rankings obtained through our information-theoretic measures (which do no require labeled data) and the generalization ability of the model on a labeled held-out set.

We consider the same tasks and networks as discussed in the prior sections, and compute the rankings using (i) extrinsic evaluation metrics defined for the task (such as validation accuracy for `Colored MNIST` and compression for `Bias-in-Bios`), (ii) the mean of entropy values, and (iii) the mean of

---

[6]Although MI values remain non-negative throughout, the x-axis in our density plots might show negative values as an artifact of fitting a Gaussian mixture model.

Table 2: We measure the correlation (Kendall's $\tau$) between model rankings based on their generalization as estimated through extrinsic metrics on labeled test sets and those obtained via information measures. Note that $\tau$ can range from -1.0 (perfect disagreement) to 1.0 (perfect agreement).

| | Sentiment Adjectives | Colored MNIST | Bias-in-Bios | | | Shuffled IMDb | Shuffled MNIST |
|---|---|---|---|---|---|---|---|
| | Validation Accuracy | Validation Accuracy | Comp-ression | TPR Gap | Suff. Gap | Validation Accuracy | Validation Accuracy |
| Mean Entropy | 0.80 | 1.00 | 0.47 | 0.20 | 0.20 | 0.60 | 1.00 |
| Mean MI | 0.80 | 1.00 | 0.60 | 0.07 | 0.33 | 0.80 | 1.00 |

MI values computed for the same networks. We then compute the Kendall rank correlation coefficient, $\tau$, between these rankings (between (i) & (ii), and (i) & (iii)) to evaluate the agreement amongst them.

We observe high correlation values for all the comparisons (Table 2). Particularly high correlations are observed for setups with synthetically induced spurious correlations (§3.1) and shuffled labels (§4), with rankings on `Colored MNIST` being perfectly correlated. Correlations on `Bias-in-Bios` are positive but lower, likely due to the more nuanced setup, where the memorization is less pronounced and extrinsic metrics are weakly correlated even among themselves (appendix D.1; Orgad et al., 2022). These positive correlations are important because—unlike the other metrics across which the correlations are computed—the information measures are purely intrinsic to the model and do not assume access to any OOD data. We perform an additional comparative discussion with standard conventional methods for model selection in appendix D.2.

## 6   Related Work

A large body of work aims at measuring and quantifying generalization, especially in out-of-distribution (OOD) scenarios (Ben-David et al., 2010; Hendrycks et al., 2021a; Wang et al., 2021b). The most common approach is to curate and label a set of examples to evaluate if networks exploit certain heuristics or shortcuts (Lapuschkin et al., 2019; Zhao et al., 2018). Several past studies create such sets spanning different domains and tasks to shed light on common failure modes in both the trained models, and the datasets used to train them. A body of such work exists for several tasks across vision (Russakovsky et al., 2015; Hendrycks & Dietterich, 2019; Hendrycks et al., 2021a,b) and NLP (McCoy et al., 2019; Naik et al., 2018; Ravichander et al., 2021; Kim & Linzen, 2020).

Closely related to the motivations for our work, past work has attempted to evaluate models using techniques that go beyond extrinsic evaluation. Training dynamics have been explored to assess the role of individual examples in training sets (Swayamdipta et al., 2020) and how specific knowledge features are temporally picked during training (Saphra & Lopez, 2019). Recent work has noted that frequent spurious artifacts are learnt prior to general patterns during training (Tänzer et al., 2021), in turn followed by memorization of individual examples (Arpit et al., 2017). Closely sharing our motivations of using information-theoretic viewpoints for intrinsic evaluation over network activations, past work has investigated *probing* or *diagnostic* classifiers (Ettinger et al., 2016; Adi et al., 2017; Belinkov et al., 2017; Hupkes et al., 2018). Researchers have further extended this paradigm to analyze the role of individual neurons (Dalvi et al., 2019; Durrani et al., 2020; Bau et al., 2017, 2020), although this approach may fail to identify causal roles (Antverg & Belinkov, 2022; Belinkov, 2022). Other work using information theory to study neural networks has focused on their learning process through the lens of the information bottleneck principle (Tishby et al., 1999), categorizing learning into distinct phases (Shwartz-Ziv & Tishby, 2017; Saxe et al., 2018) and obtaining generalization bounds (Tishby & Zaslavsky, 2015). Follow-up work has made use of such measures to regularize training for robustness (Wang et al., 2021a) and low-resource learning (Mahabadi et al., 2021).

## 7   Limitations and Future Directions

Below, we describe some of the limitations of our work and discuss future research directions.

**Comparative Nature of Observations**   The current findings and insights derived from the information measures are comparative in nature that could be a limitation when being applied for practical use cases. In order to assess some given models using the information measures described herein, we must a-priori know at least one of two things: (i) one of the given models that generalizes well, so that the rest could be bench-marked against it, or (ii) the kind of memorization that the models trained on the dataset are expected to possess, so that we could make a comparison between the given models. Further, one may want to compare models that do not belong to the same model family, architecture and hyperparameter set. In such cases, values from our information measures might not be directly comparable across these different models. We design a simple experiment to study this in appendix C.2 where we compute our measures for models with varying capacity. We note that values for networks with different capacities lie on different scales and hence are not directly comparable.

**Scaling to Larger Models**   While the analysis presented in this work is performed for small to moderately large networks like MLPs, RoBERTa, and ResNet—for whom our hypothesized trend holds consistently—more research is needed to study the scaling behavior of these information measures as a function of data and model size (Rosenfeld et al., 2020; Kaplan et al., 2020).

**Practical Applications.**   In this work, we show the utility of our observations for the preliminary use case of model selection. More research is required to investigate the usefulness of our observations in other scenarios. One viable direction to explore is the problem of *OOD detection* (Arora et al., 2021)—deciding whether a specific data point is OOD—by computing point-wise versions of the information measures. Another case where the proposed information measures could also be useful is *regularizing models*, where regularizing the MI/entropy values to a certain a band of values might yield more generalizing models. Such regularization can also be coupled with our understanding of training dynamics from prior work (Tänzer et al., 2021) that has identified training stages where particular forms of memorization is seen to exist. Our understanding of *training dynamics*, in itself, could be enhanced by studying the progression of neuron diversity across training steps and hence noting the generalization patterns that emerge.

## 8   Conclusion

In this work, we have taken a step towards identifying generalization behavior of neural network models based on their intrinsic activation patterns. We presented information-theoretic measures that allow us to distinguish between models that show two kinds of memorization: those that pick up surface-level spurious correlations (heuristic memorization) and those that overfit on individual training instances (example-level memorization). Through investigations spanning multiple natural language and vision tasks, we corroborated our hypothesis that such memorization is reflected in diversity across neural activations, and hence the defined information measures that quantify them. Finally, we demonstrated a potential application of this framework for model selection.

## Acknowledgments

We are grateful to the Technion CS NLP group and others at the Technion—particularly, Mor Ventura, Michael Toker, Hadas Orgad, Reda Igbaria, Zach Bamberger, Adir Rahamim, Anja Reusch, and Gail Weiss—for the insightful discussions that shaped this work. RB would like to extend his gratitude to his dorm-mates and friends—Atulya, Josh, Cornelius, David, Kristóf, Pratibha, Ajay, Navdeep—for being a constant source of home during his time at the Technion. RB would also like to thank the support staff at Delhi Technological University and the Technion for their administrative support. We also thank the anonymous reviewers and area chairs during the review process at NeurIPS 2022 for their careful analysis of our work. This research was supported by the Israel Science Foundation (grant No. 448/20) and by an Azrieli Foundation Early Career Faculty Fellowship.

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
