# A Computing Information Measures

Our goal is to compute the Entropy for a neuron, $A_x$, and MI between a pair of neurons, $A_x$ and $A_y$, for some $(x, y) \in \{1, \ldots, N\}$, where the neurons are represented as their activations for a given number of samples (S).

## A.1 Entropy

---

**Algorithm 2** Computing entropy

---

1: **procedure** ENTROPY($A_x$)
2:     $H(A_x) \leftarrow 0$
3:     $\hat{A}_x \leftarrow \text{BIN}(A(x, i))$                                                ▷ Discretizing activations by binning
4:     **for** $i \in \{1, \ldots, N_{bins}\}$ **do**
5:         $p(\hat{a}_{(x,i)}) \leftarrow \{\mathbb{1}\{\hat{a}_{(x,i)} == \hat{a}_{(x,j)}\}\}_{j=1}^{S} / S$                     ▷ Computing probability
6:         $h(a_{(x,i)}) \leftarrow p(\hat{a}_{(x,i)}) \log(1/p(\hat{a}_{(x,i)}))$                          ▷ As per Eq. 1
7:         $H(A_x) \leftarrow H(A_x) + h(a_{(x,i)})$
8:     **end for**
9:     $H(A_x) \leftarrow H(A_x) / S$                                              ▷ As per Eq. 1
10:    **return** $H(A_x)$
11: **end procedure**

---

In order to compute entropy, we first discretize the set of neuron activations by binning them in a uniform range of values (Darbellay & Vajda, 1999). Specifically, we divide the range of activations for a neurons into a constant number of bins (usually 100), each of the same size. Then, the probability for each activation is determined by the number of activations that share the same discrete value. Plugging these values in Equation 1, across all activations for a neurons gives us the entropy for that neuron.

## A.2 Mutual Information

To compute the MI between these one-dimensional variables, we use the estimator proposed in Kraskov et al. (2004), which provides a tight lower bound to the mutual information, especially in low-dimensional cases (Belghazi et al., 2018).

Kraskov et al. (2004) consider the popular interpretation of mutual information $I(X; Y)$ between two continuous random variables $X$ and $Y$ as "the amount of uncertainty left in $Y$ when $X$ is known". In terms of Shannon entropy, this is equivalent to $I(X; Y) = H(Y) - H(Y|X) \equiv H(X) + H(Y) - H(X, Y)$. Following this formalization, we require to work with three variables- $A_x$, $A_y$, and $A_z = (A_x, A_y)$. To compute the MI, the $k$-nearest neighbor distances are considered for these variables. Particularly, for some given distance metric, we represent $\epsilon_i$ as the distance of a given point $a_{(z,i)} = (a_{(x,i)}, a_{(y,i)})$ to its $k^{th}$-neighbor. This distance is then considered in the $x$ and $y$ spaces—$e_{(x,i)}$ and $e_{(y,i)}$ depict the number of points that lie at a distance lesser than $\epsilon_i$ with respect to $a_{(x,i)}$ and $a_{(y,i)}$, respectively. Then, the mutual information is given as:

$$I(A_x; A_y) = \psi(k) + \psi(S) - \frac{1}{S} \sum_{i=1}^{S} (\psi(e_{(x,i)}) + \psi(e_{(y,i)})) \tag{3}$$

where, $\psi(\cdot)$ is the digamma function, or the logarithmic derivative of the gamma function $\Gamma(\cdot)$: $\psi(x) = \frac{\Gamma'(x)}{\Gamma(x)} = \ln x - \frac{1}{2x}$.

We use the Chebyshev distance or the $L_\infty$ metric as our distance metric for all distance computations in the $X, Y$, and $Z$ spaces.

Popular recent estimators take a neural approach for estimating MI and consider an alternate interpretation of MI as the dependence between two random variables, i.e., $I(X; Y) = D_{KL}(P_{(X,Y)} || P_X \otimes P_Y) = D_{KL}(P_{(Y|X)} || P_Y)$. Since $p(y|x)$ is a-priori unknown for most real-world distributions, Cheng et al. (2020) approximate it through a parametric variational distribution $q_\theta(y|x)$. Belghazi et al. (2018) instead consider representations that estimate the KL divergence. They parameterize the

**Algorithm 3** Computing mutual information

---

1: **procedure** MI$(A_x, A_y)$
2:      $I(A_x; A_y) \leftarrow \psi(k) + \psi(S)$                                                    ▷ As per Eq. 3
3:      $A_z \leftarrow A_x \bigoplus A_y$
4:      KNN $\leftarrow$ KNN$(A_z)$                                     ▷ Initializing k-nearest neighbor distances
5:      $e_x \leftarrow 0, e_y \leftarrow 0$
6:      **for** $i \in \{1, \ldots, S\}$ **do**
7:          $\epsilon_i \leftarrow$ KNN$(a_{(z,i)}).r(k)$                                          ▷ Distance to the $k^{th}$ neighbor
8:          $e_{(x,i)} \leftarrow \{\mathbb{1}\{||x_j - x_i|| < \epsilon_i\}\}_{j=1}^{X-i}$
9:          $e_{(y,i)} \leftarrow \{\mathbb{1}\{||y_j - y_i|| < \epsilon_i\}\}_{j=1}^{X-i}$
10:         $e_x \leftarrow e_x + \psi(e_{(x,i)}), e_y \leftarrow e_y + \psi(e_{(y,i)})$
11:     **end for**
12:     $I(A_x; A_y) \leftarrow I(A_x; A_y) - \frac{1}{|X|}(e_x + e_y)$                                ▷ As per Eq. 3
13:     **return** $I(A_x; A_y)$
14: **end procedure**

---

family of functions that defines the bound of these representations using a neural network. However, these approaches require us to learn a distinct neural network for each pair of random variable across which we wish to compute the MI. Considering the large number of pairwise operations, especially when working at the level of individual neurons, these approaches are not feasible for our analysis.

# B    Extended Experimental Details

Table 3: Values for network and training hyper-parameters across different experimental settings.

| Model | # Layers | # Parameters | # Epochs | LR | Batch size | Pretrained? |
|---|---|---|---|---|---|---|
| MLP (Colored MNIST) | 3 | 0.232M | 50 | 0.001 | 512 | ✗ |
| MLP (Shuffled MNIST) | 3 | 0.184M | 200 | 0.001 | 512 | ✗ |
| DistilBERT | 6 | 66M | 20 | 5e-5 | 64 | ✓ |
| RoBERTa-base | 12 | 123M | 10 | 5e-5 | 64 | ✓ |
| ResNet-18 | 18 | 11M | 25 | 1e-5 | 64 | ✗ |

Details for network architectures and training regimes are given in Table 3. Further explanations regarding datasets for the various experimental settings are elaborated below.

## B.1    Colored MNIST

All images in the training and evaluation sets are colored green or red. $\alpha \in \{0.00, 0.25, 0.50, 0.75, 1.00\}$ corresponds to the fraction of training examples that follow the color-to-label correlation: green $\rightarrow$ label-0 (digits 0–4) and red $\rightarrow$ label-1 (digits 5–9). The size of the set is same as the original MNIST set, that is a training set of 50k examples and an evaluation set of 10k examples. The hyper-parameters for model training were picked from (Arjovsky et al., 2019).

## B.2    Sentiment Adjectives

The chosen set of adjectives for sub-sampling examples from IMDb and then induce the adjective-based spurious correlations are: **Positive**: {*good*, *great*, *wonderful*, *excellent*, *best*}, and **Negative**: {*bad*, *terrible*, *awful*, *poor*, *negative*}. To maintain the correlation for each individual adjective, the set does not contains examples that contain adjectives from both of the sets. The size of this sampled set came to be $\sim$6k, out of which 67% of the example already abide to the heuristic. That is, the true label corresponding to these examples is same as the nature of adjective(s) they contain. Thus, we vary the labels corresponding to the other 33% of the examples with increasing $\alpha$. Particularly, $\alpha$ takes a value in $\{0.00, 0.25, 0.50, 0.75, 1.00\}$, where 0.00 suggests that 33% of the data does not follow the heuristic, while 1.00 suggests that all examples follow the heuristic. The learning rate

(LR) and batch size was chosen after a hyper-parameter sweep with values $\{5e-3, 3e-4, 5e-5\}$, $\{16, 32, 64\}$ on the generalizing training set ($\alpha = \beta = 0.00$).

### B.3 Bias-in-Bios

We perform our analysis over trained models released by Orgad et al. (2022) for randomly picked random seeds of 0, 5, 26, 42, and 63. `Bias-in-Bios` (De-Arteaga et al., 2019) contains around 400k biographies, the networks are trained on $65\%$ of this set, while a sub-sampled balancing from the rest of examples are used for evaluation and analysis.

### B.4 NICO++

We sub-sample 6 animal classes with the most number of examples across them (*bear*, *dog*, *cat*, *bird*, *horse*, *sheep*) from the original dataset, spanning ∼10k training and ∼7.8k evaluation examples. Both `balanced` and `unbalanced` set contain all individual contexts for the animals. For the `unbalanced` set: one from ten common contexts is associated to each of the chosen animal classes (*grass*, *water*, *autumn*, *dim*, *outdoor*, *rock*, respectively). On the other hand, the `balanced` set contains equal number of images from all these contexts for each of the animal classes.

### B.5 Shuffled MNIST

Labels are shuffled for the 10 digits of MNIST over the 50k training examples. $\beta \in \{0.00, 0.25, 0.50, 0.75, 1.00\}$. The evaluation and analysis is performed over 10k balanced testing examples from the original set.

### B.6 Shuffled IMDb

The networks are trained for the 25k training examples and shuffled for $\beta \in \{0.00, 0.25, 0.50, 0.75, 1.00\}$. Performance computations and analysis is performed on 5k examples sampled from the evaluation set.

## C Extended Experimental Results

### C.1 Semi-synthetic Heuristic Memorization

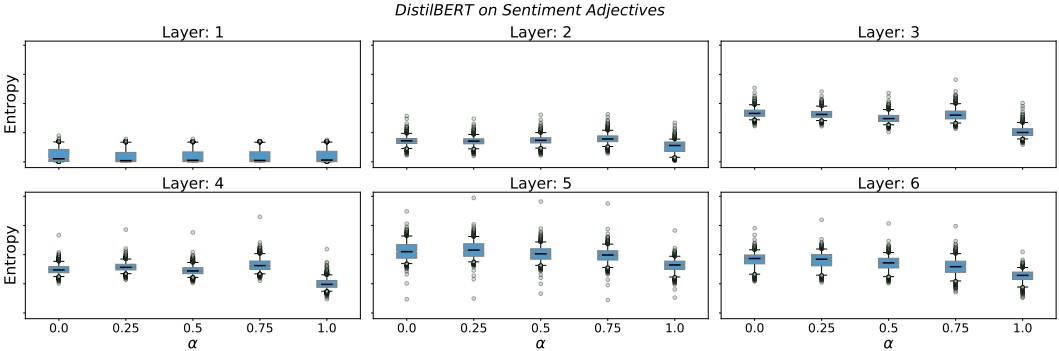

Figure 8: Variation of neuron entropy across all layers of DistilBERT on `Sentiment Adjectives`.

Figure 8 shows the variation of neuron entropy across increasing values of $\alpha$ (here, adjective-to-sentiment correlation during training) for all layers of DistilBERT. As discussed in the main text (§3.1), we primarily see a variation in entropy for later layers in the network. In the first layer, we observe that entropy for all values of $\alpha$ remains similar and low, while as we go towards the later layers, we start to see certain differences in entropy, with the network for $alpha = 1.0$ showing especially lower entropy. Since all these networks have been fine-tuned from the same pre-trained initialization and vary only in terms of how much spurious correlation is present in their training sets, we attribute this pattern across layers to a possibility that such spurious correlation are largely captured in the later layers during fine-tuning.

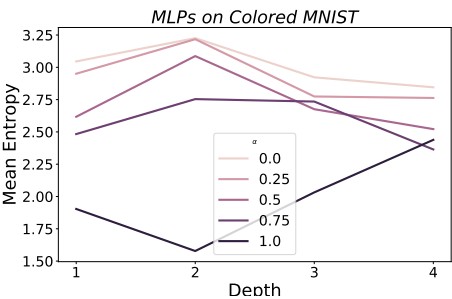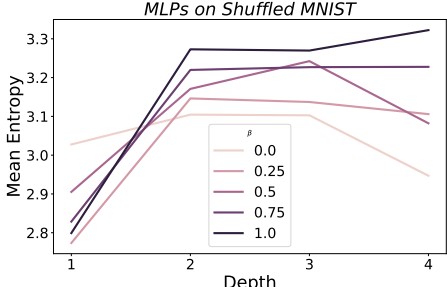

Figure 9: Comparison of entropy for MLP networks with varying number of layers.

## C.2 Effect of Model Capacity

Theoretically, we expect that capacity of a network would influence entropy and MI. To test this, we train MLP networks with varying depths for the tasks of Colored MNIST and Shuffled MNIST, and compare the neuron entropy for the obtained networks. From results shown in Figure 9 observe that our hypothesis holds true for all capacities (ranging from single to four-layered networks). However, different capacity networks show entropy values in different ranges and hence they are likely not directly comparable with each other.

## C.3 Bias-in-Bios variation across layers

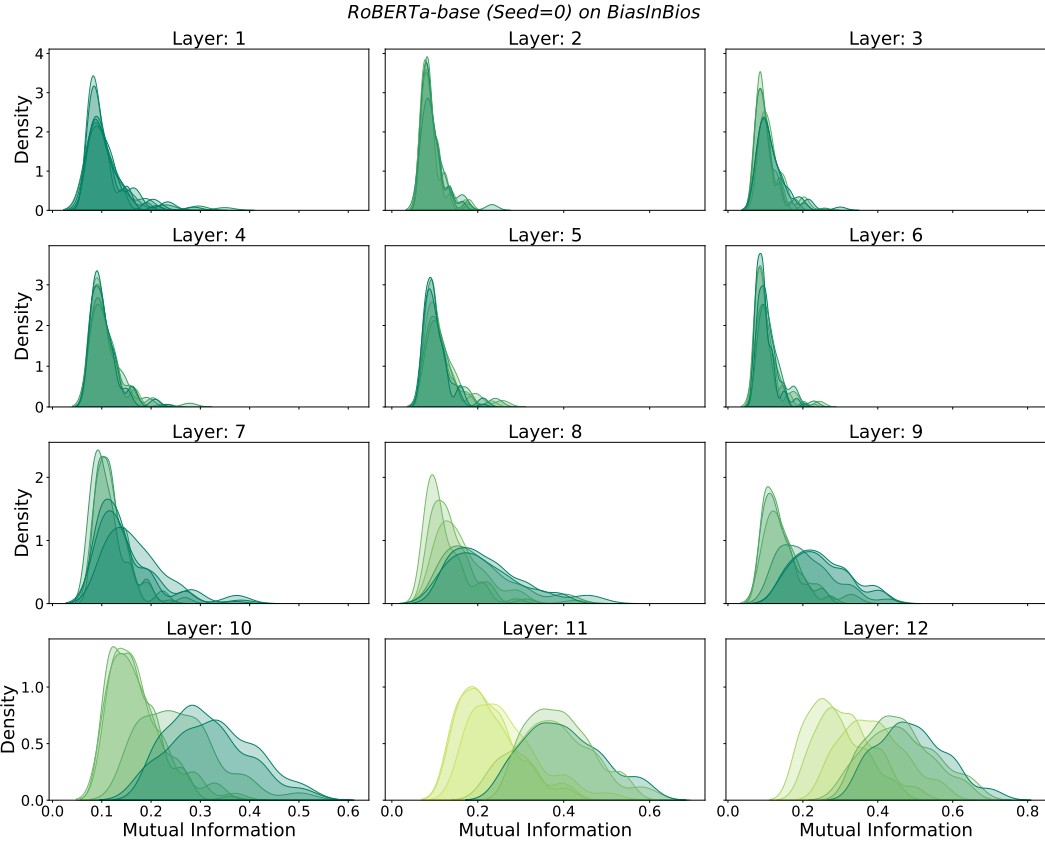

Figure 10: Distribution of MI across neuron pairs for different layers in RoBERTa-base fine-tuned on `Bias-in-Bios` training sets. Colors represent the value of gender extractability with the MDL probe.

Figure 10 shows how the density of neuron pairs across MI values changes across layers of RoBERTa-base when trained on different training sets of `Bias-in-Bios`. By looking at these distributions of MI in conjunction with the value of compression (obtained through the MDL probe), we observe a very interesting phenomenon: The relation between MI and compression is maintained throughout the network across its different layers. Networks trained on the different training sets have very similar presence of gender information, as indicated by compression values (color of the Gaussian plots). This is replicated in the MI distribution where all plots are clustered at very similar values. We see that the distinction among the different sets stars to become more lucid with later layers (reflected in the colors) and the same distinction is observed with MI, where more biased models (darker colors) show higher MI. We reported the results for the last layer in the main text (§3.2) because compression values vary the most at this layer, and is consequently reflected in MI. However, looking at the variation across all layers here validates the usage of information measures as a tool for evaluation even strongly.

### C.4 Relation to Complexity Measures

Table 4: Pearson correlation coefficient (PCC) between norm-based complexity measures and information measures for the two memorization types.

| Complexity Measures | PCC for Colored MNIST | | PCC for Shuffled MNIST | |
|---|---|---|---|---|
| | **Mean Entropy** | **Mean MI** | **Mean Entropy** | **Mean MI** |
| 2-Norm | 0.31 | 0.14 | 0.96 | 1 |
| Frobenius-Norm | 0.5 | 0.31 | 0.99 | 0.95 |
| Path-Norm | 0.58 | 0.3 | 0.98 | 0.94 |
| Validation Acc. | 0.96 | 0.85 | 0.87 | 0.7 |

Here, we validate that the activation diversity does not simply capture the complexity of a learned model. Popular norm-based measures are often used as complexity measures (also used for model selection; see appendix D.2). On computing the absolute pearson correlation between these complexity measures and our proposed information measures in Table 4, we observe that the two are weakly correlated for `Colored MNIST` and strongly correlated for `Shuffled MNIST`. This suggests that the information measures might approximate to model complexity in the case of example-level memorization, but captures more than just complexity as evident for heuristic memorization. Notably, in both cases, our measures are strongly correlated to the validation accuracy.

### C.5 Variance of Entropy and MI

As seen from results on our various experimental setups (Figures 1-7), typically a wide distribution of values is observed for both entropy and MI, for all models trained on different $\alpha$ and $\beta$ values. Since entropy is computed for each neuron in the network and MI for all pairs of neurons, a large number of values are obtained for each network, resulting to this wide distribution. Even though these distributions have a wide range, an observable difference is seen that follows the expected trend across both kinds of memorizations. This is further apparent through the model selection results where even the mean over these two distributions yields encouraging rankings. However, such a variance could make it challenging for practitioners to identify differences in cases when the amount of memorization between two models is not high (for instance, a difference of $\alpha = 0.5$).

## D Extended Analysis for Model Selection

### D.1 Model Rankings for Bias-in-Bios

Here, we compute the Kendall $\tau$ ranking correlations between all ranking metrics for the `Bias-in-Bios` dataset (Table 5). We observe that extrinsic metrics (TPR, FPR, Separation, and Sufficiency Gap) share higher correlations between each other than those with MI and Entropy, that are only weakly positive. All these extrinsic metrics are computed by training and assessing the networks on specifically curated sets that are labeled for gender information and are thus expected to be more similar to each other. However, we see that compression (an intrinsic metric that still requires labeled data) correlates in a similar manner to extrinsic as well as the information theoretic measures (not requiring labeled data).

Table 5: Kendall's $\tau$ between model rankings from all pairs of intrinsic and extrinsic metrics. $\tau$ can range from -1.0 (perfect disagreement) to 1.0 (perfect agreement).

|  | Comp-ression | TPR Gap | FPR Gap | Separation Gap | Sufficiency Gap | Entropy (Mean) | MI (Mean) |
|---|---|---|---|---|---|---|---|
| Compression | 1.00 | 0.47 | 0.60 | 0.47 | 0.47 | 0.47 | 0.60 |
| TPR Gap | 0.47 | 1.00 | 0.87 | 0.73 | 0.73 | 0.20 | 0.07 |
| FPR Gap | 0.60 | 0.87 | 1.00 | 0.60 | 0.60 | 0.07 | 0.20 |
| Separation Gap | 0.47 | 0.73 | 0.60 | 1.00 | 1.00 | 0.20 | 0.33 |
| Sufficiency Gap | 0.47 | 0.73 | 0.60 | 1.00 | 1.00 | 0.20 | 0.33 |
| Entropy (Mean) | 0.47 | 0.20 | 0.07 | 0.20 | 0.20 | 1.00 | 0.60 |
| MI (Mean) | 0.60 | 0.07 | 0.20 | 0.33 | 0.33 | 0.60 | 1.00 |

## D.2 Comparison with Baseline Generalization Measures

We perform model selection on several benchmark generalization measures. Norm-based measures are popularly used in literature as a measure of generalization (Jiang et al., 2019), and we conduct comparisons with three of them: 2-norm, Frobenuis-norm, and Path-norm. We do this for all our setups across example-level and heuristic memorization (Path-norm is not computed on the transformer models, since no established way exists to do so). On comparing the model selection performance with our information measures, we observe that while these baselines (specifically Frobenius-norm and Path-norm) do reasonably well in ranking models that perform example-level memorization, they do poorly for heuristic memorization. This follows from the fact that these metrics are conventionally based and tested on the more widely known notion of memorization, that of individual examples. Where 2-norm gives $\tau = 1$ on Shuffled-IMDb, it is totally uncorrelated ($\tau = 0.00$) for Sentiment Adjectives.

At the same time, our information measures hold positive $\tau$ values across all models and tasks, while the baseline measures are inconsistent, varying from extremely low to high values across setups. Frobenius-norm shows perfect negative and positive correlations with Shuffled IMDb and MNIST, respectively, depicting that a high norm value could mean anything depending on the dataset and model.

Table 6: Expanded results for model selection with added Kendall $\tau$ ranking correlations for norm-based generalization measures (Jiang et al., 2019).

|  | Sentiment Adjectives | Colored MNIST | Bias-in-Bios | | | Shuffled IMDb | Shuffled MNIST |
|---|---|---|---|---|---|---|---|
|  | Validation Accuracy | Validation Accuracy | Comp-ression | TPR Gap | Suff. Gap | Validation Accuracy | Validation Accuracy |
| Mean Entropy | 0.80 | 1.00 | 0.47 | 0.20 | 0.20 | 0.60 | 1.00 |
| Mean MI | 0.80 | 1.00 | 0.60 | 0.07 | 0.33 | 0.80 | 1.00 |
| 2-Norm | 0.00 | -0.20 | 0.33 | 0.60 | 0.87 | 1.00 | 0.80 |
| Frobenius-Norm | -0.40 | 0.80 | -0.20 | 0.07 | 0.07 | -1.00 | 1.00 |
| Path-Norm |  | 0.80 |  |  |  |  | 1.00 |

# E  Compute Resources and Details

For our experiments that work with MLP networks, we use a single NVIDIA GeForce RTX 2080 GPU (16 GB) to perform the training and inference. For other experiments with larger transformer and convolutional networks—DistilBERT, RoBERTa, and ResNet-18—we use a single NVIDIA A40 GPU (48 GB) to perform fine-tuning/training and inference. We use pre-trained checkpoints from prior work whenever possible and limit the amount of training we do ourselves. In all, we use ~190 GPU hours for our experiments, a breakdown of which is given in Table 7.

Table 7: GPU hours used per experiment.

| | # Types (Dataset/Training Configurations) | # Seeds/ Type | Training | | Inference | | Total Time (Hours) |
|---|---|---|---|---|---|---|---|
| | | | Time/Run (Hours) | Total (Hours) | Time/Run (Hours) | Total (Hours) | |
| MLP (Colored MNIST) | 5 | 10 | 0.25 | 12.5 | 0.05 | 2.5 | 15 |
| MLP (Shuffled MNIST) | 5 | 10 | 1 | 50 | 0.05 | 2.5 | 52.5 |
| DistilBERT (Sentiment Adjectives) | 5 | 5 | 1 | 25 | 0.20 | 5 | 30 |
| DistilBERT (Shuffled IMDb) | 5 | 5 | 2.5 | 62.5 | 0.20 | 5 | 67.5 |
| RoBERTa-base | 5 | 5 | - | - | 0.5 | 12.5 | 12.5 |
| ResNet-18 | 2 | 5 | 1 | 10 | 0.25 | 2.5 | 12.5 |
| **Total GPU Time** | | | | | | | **190** |

# F    Potential Societal Impacts

Our work in aimed at comparing a given set of models by shedding light on their potential reliance on two facets of memorization. While such evaluation is meant to be used to pick the best model, (for instance, the least biased one) one could instead use it to choose the least generalizing model (the most biased). This can be especially worrisome in sensitive scenarios like that of gender bias, where one could easily switch the order of entropy and MI to obtain a network which attends to gender information the most.

# G    Dataset Licenses

All datasets used in this work are freely publicly available.

**Bias-in-Bios:** MIT License

**MNIST:** GNU General Public License v3.0

**IMDb:** License statement can be found here.

**NICO:** License statement can be found here.