# OpenReview forum: "Measures of Information Reflect Memorization Patterns"
_NeurIPS.cc/2022/Conference — NeurIPS 2022 Accept_

### Official Review · Reviewer_yJeZ · 2022-07-06

**Rating:** 4
**Confidence:** 4
**Soundness:** 3 good
**Presentation:** 4 excellent
**Contribution:** 1 poor

**Summary:**

This paper studies two types of memorization: i) heuristic memorization and ii) example-level memorization. The authors study the diversity of activations units of the neural networks either across a set of examples (intra-neuron) or for the same example (inter-neuron). They observe, through various experiments on both synthetic and real datasets, that models experiencing heuristic memorization have lower diversity in their activations compared to models experiencing example-level memorization. They also suggest using their two metrics for model selection.

**Questions:**

1- In Figure 1, how would you compare the generalization ability of the three networks that are trained on the three tasks? Would you assume that the first model (trained on the base task) is the most "generalizable"? If so, then how would you explain Figure 1 (g) and (h) in terms of capturing generalization? Ranking the models based on these two figures gives us: model (b) > model (a) > model (c). Hence, if Neuron Entropy and Density of Neuron Pairs are reflective of generalization, this result would suggest that the model with "heuristic memorization" is the most "generalizable", which is counterintuitive.

2- The paper tries to distinguish heuristic memorization from example-level memorization. But what is not clear is how to distinguish memorization (either type) from no memorization? If the method is not able to do so, then I would suggest combining this approach with a method that detects memorization. That is, first, detect with that method whether the model is experiencing any memorization or not. Next, if the model is memorizing, then distinguish its type using your approach (low/high entropy/MI). This would be a much more interesting tool.

3- One concern raised in the introduction is that it's difficult to generate OOD datasets. How is this solved in your paper?

4- Why are the activation units of only the encoder considered? Why not use both the encoder and the decoder? What is even meant here by encoder and decoder? (lines 58-63)

5- How do you make sure that the models trained on synthetic datasets with heuristic memorization do actually memorize? This is important to double-check. Because it could be that the network chooses to classify based on the more complex features (for example the circles in Figure 1) rather than based on the simple induced feature. If that's the case, then studying that model is not the same as studying heuristic memorization. The validation accuracy shows this to some extent, but it would be better to see some examples as well. For example, for a new green point in the outer circle but high in the induced feature, what is the label?

**Limitations:**

The authors have not mentioned the limitations of their method in the main paper. But in the appendix, they mention potential societal impacts.

**Strengths And Weaknesses:**

Strengths:

1- The paper is well-written and the figures look nice.

2- To the best of my knowledge, the proposed metrics to distinguish between two types of memorization are novel.

3- The empirical results include both vision and language tasks, and also both synthetic and real datasets.

Weaknesses:

1- The application for the method is not clear. It is also not clear what are the particular insights that this study brings to the table; is it that "models experiencing example-level memorization are more complicated than models experiencing heuristic memorization", and that "models experiencing heuristic memorization are less complicated than models not experiencing that"? Aren't these conclusions expected from our understandings of neural networks? What is the new outcome of this study?

2- Comparison with other complexity measures is missing. If the activation diversity is simply capturing the complexity of the learned model, then how does it compare to other neural network complexity measures? What would be the specific advantage of these new metrics?

3- Lack of comparison to other methods also applies to the model selection section. How does the model compare to other generalization measures? Also, the obtained Kendall tau is computed for which models? What is the scope of this model selection? If it's just computed over a few networks then the generalizability would be questionable.

---

> ### Author Response · Authors · 2022-08-02
> **Response to Reviewer yJeZ (1/3)**
>
> We thank the reviewer for their careful analysis of our work, and for their insightful comments. We are encouraged to see that the reviewer found our presentation and visualizations apt in capturing our results and hypotheses. We are glad that they also acknowledge the novelty of our approach and the comprehensiveness of our experiments.
>
> We recognize that one of their biggest concerns is about potential applications:
>
> > The application for the method is not clear. It is also not clear what are the particular insights that this study brings to the table; is it that "models experiencing example-level memorization are more complicated than models experiencing heuristic memorization", and that "models experiencing heuristic memorization are less complicated than models not experiencing that"? Aren't these conclusions expected from our understandings of neural networks? What is the new outcome of this study?
>
> First, some of these insights (and trends) are easy to explain _in hindsight_, but before conducting the study we couldn’t guess them, and weren’t sure if they would consistently hold across different experimental conditions. Further, all of our analysis is performed on in-distribution input samples, yet it provides insights about out-of-distribution performance.
>
> Regarding applications: In the paper, we show how some of our insights could be applied to the problem of model selection. Further, we believe that our analysis holds potential for impacting other applications, e.g., improving the model generalization through regularization, OOD detection etc. (Please also see the general response and responses to Reviewer hktY for details on these applications).
>
> >If the activation diversity is simply capturing the complexity of the learned model, then how does it compare to other neural network complexity measures? What would be the specific advantage of these new metrics?
>
> No, the activation diversity does not simply capture the complexity of the learned model. The two diversity measures allow us to study the behaviour of individual neurons across examples and shared information between different neurons. Norm-based measures that we have used to benchmark our model selection experiments (and also suggested by Reviewer hktY) are also often used as complexity measures. On computing the absolute pearson correlation between these complexity measures and our proposed information measure, we observe that our measures correlate better with heuristic memorization than the baseline measures (which are uncorrelated).  For correlation with example-level memorization, both the baseline measures and our measures correlate to a large extent. In both cases, our measures are strongly correlated to the validation accuracy, which suggests that the information measures capture more than just the complexity of a network.
>
>
> Please see the results below and the response to the next comment.
>
>
>
> | Corr. for Colored MNIST | Mean Entropy | Mean MI |
> |-------------------------|--------------|---------|
> | 2-Norm                  |        0.31 |    0.14 |
> | Frobenius-Norm          |        0.50 |    0.31 |
> | Path-Norm               |        0.58 |    0.30 |
> | Validation Acc.          |         0.96 |   0.85 |
>
> The table above captures heuristic memorization.
>
> | Corr. for Shuffled MNIST | Mean Entropy | Mean MI |
> |--------------------------|--------------|---------|
> | 2-Norm                   |         0.96 |   1.00 |
> | Frobenius-Norm           |         0.99 |   0.95 |
> | Path-Norm                |         0.98 |    0.94 |
> | Validation Acc.          |         0.87 |   0.70 |
>
> The table above captures example-level memorization.

---

> > ### Author Response · Authors · 2022-08-02
> > **Response to Reviewer yJeZ (2/3)**
> >
> > > How does the model compare to other generalization measures? Also, the obtained Kendall tau is computed for which models? What is the scope of this model selection?
> >
> > We have now performed model selection on several benchmark generalization measures, as suggested by the reviewer. Norm-based measures are popularly used in literature as a measure of generalization, and we conduct comparisons with three of them: 2-norm, Frobenuis-norm, and Path-norm. We do this for all our setups across example-level and heuristic memorization (Path-norm is not computed on the transformer models, since no established way exists to do so). On comparing the model selection performance with our information measures, we observe the following:
> >
> > While these baselines (specifically Frobenius-norm and Path-norm) do  reasonably well in ranking models that perform example-level memorization, they do poorly for heuristic memorization. This follows from the fact that these metrics are conventionally based and tested on the more widely known notion of memorization, that of individual examples. Where 2-norm gives \tau=1 on Shuffled-IMDb, it is totally uncorrelated (\tau=0.00) for Sentiment Adjectives.
> > Our information measures hold positive \tau values across all models and tasks, while the baseline measures are inconsistent, varying from extremely low to high values across setups. Frobenius-norm shows perfect negative and positive correlations with Shuffled IMDb and MNIST, respectively, depicting that a high norm value could mean anything depending on the dataset and model.
> >
> > |                      |                 | Sentiment  Adjectives |    Colored  MNIST   |             | Bias-in-Bios |                 |   Shuffled  MNIST   |    Shuffled  IMDb   |
> > |----------------------|-----------------|:---------------------:|:-------------------:|:-----------:|:------------:|:---------------:|:-------------------:|:-------------------:|
> > |                      |                 |  Validation Accuracy  | Validation Accuracy | Compression |    TPR-Gap   | Sufficiency-Gap | Validation Accuracy | Validation Accuracy |
> > | Information Measures | Mean of Entropy |                  0.80 |                1.00 |        0.47 |         0.20 |            0.20 |                1.00 |                0.60 |
> > |                      |    Mean of MI   |                  0.80 |                1.00 |        0.60 |         0.07 |            0.33 |                1.00 |                0.80 |
> > |       Baselines      |      2-Norm     |                  0.00 |               -0.20 |        0.33 |         0.60 |            0.87 |                0.80 |                1.00 |
> > |                      |  Frobenius-Norm |                 -0.40 |                0.80 |       -0.20 |         0.07 |            0.07 |                1.00 |               -1.00 |
> > |                      |    Path-Norm    |                       |                0.80 |             |              |                 |                1.00 |                     |
> >
> >
> >
> > The model selection is performed on the same models considered in the paper's prior sections. That is, for a setup like that of Coloured-MNIST, we consider the models across different values of \alpha as analyzed in Section 3.1. The Kendall tau correlation coefficient is computed between rankings obtained from different measures (such as the mean of MI or mean of Entropy) with the true rankings of the models (as obtained from the validation performance).
> >
> > > In Figure 1, how would you compare the generalization ability of the three networks that are trained on the three tasks? Would you assume that the first model (trained on the base task) is the most "generalizable"? If so, then how would you explain Figure 1 (g) and (h) in terms of capturing generalization? Ranking the models based on these two figures gives us: model (b) > model (a) > model (c).
> >
> > In Figure 1, the model trained on the original concentric circles' setup (Figure 1(a)) is the generalizing case since the model trained on this dataset captures the desired circular decision boundary to separate the two classes. Figure 1(b) does not contain any discernible patterns for solving the task, and the model trained on this set falls to example-level memorization with enough training iterations. Figure 1(c), on the other hand, consists of a straightforward linear decision boundary that separates the two circles for most examples, and hence the model trained on this data memorizes this linear boundary. Hence, 1(a) is the most generalizable.
> > One of our central hypotheses is that memorisation over examples (1(b)) leads to higher entropy and lower MI, while over heuristics (1(c)) leads to lower entropy and higher MI. We see precisely this pattern through Figures 1(g) and 1(h). We do not say that high entropy or low MI is better. Our claim and evidence talk about the direction of change in the two quantities, in comparison to the generalizing case, when one form of memorization exists.

---

> > > ### Author Response · Authors · 2022-08-02
> > > **Response to Reviewer yJeZ (3/3)**
> > >
> > > > The paper tries to distinguish heuristic memorization from example-level memorization. But what is not clear is how to distinguish memorization (either type) from no memorization? If the method is not able to do so, then I would suggest combining this approach with a method that detects memorization. That is, first, detect with that method whether the model is experiencing any memorization or not. Next, if the model is memorizing, then distinguish its type using your approach (low/high entropy/MI). This would be a much more interesting tool.
> > >
> > > This is a great idea, and would be interesting to explore in the future. Presently, our framework is relative in nature. That is, for any given pair of models (without knowing their generalization rates), we can compare the diversity of their activations and draw indications of their potential generlization behviour using just in-distribution input examples.
> > >
> > > >One concern raised in the introduction is that it's difficult to generate OOD datasets. How is this solved in your paper?
> > >
> > > Our information measures are computed on in-distribution datasets. Prior to our work, an indication of OOD performance was possible only through specially curated OOD sets (which, as you quoted, are hard to generate). Since both our measures can provide indications of memorization (that impedes OOD generalization), one can potentially perform a comparison between models without requiring OOD datasets.
> > >
> > > > Why are the activation units of only the encoder considered? Why not use both the encoder and the decoder? What is even meant here by encoder and decoder? (lines 58-63)
> > >
> > > All considered experimental settings are performed on classification tasks and thus encoder-only models (such as the RoBERTa-base encoder for Bias in Bios). Here, by "decoder" we simply refer to the classification head that takes the encoder representations and maps them to the output space. We’ll update the paper to make this clear.
> > >
> > > > How do you make sure that the models trained on synthetic datasets with heuristic memorization do actually memorize? This is important to double-check. Because it could be that the network chooses to classify based on the more complex features (for example the circles in Figure 1) rather than based on the simple induced feature. If that's the case, then studying that model is not the same as studying heuristic memorization. The validation accuracy shows this to some extent, but it would be better to see some examples as well. For example, for a new green point in the outer circle but high in the induced feature, what is the label?
> > >
> > > Two models under comparison in any of our considered setups vary only in terms of the dataset on which they are trained. For instance, for Colored-MNIST, the compared models are trained on varying values of $\alpha$, which signifies the amount of correlation between the color of input images and corresponding labels. Here, we observe that a network trained on a higher $\alpha$ leads to a lower evaluation performance on the validation set (where the heuristic no longer holds). In such a case, a drop in the validation performance itself is a strong indicator of the model's memorization.
> > >
> > > For other cases where the spurious correlations are more subtle, we do perform evaluation on specified datasets and use task-specific metrics. For gender bias, we use Bias in Bios, which is a specially curated set that elucidates the sensitivity of language models on gender features. While, NICO++ is used for the task of image classification, where models tend to rely on image backgrounds more than the target object.

---

> > > > ### Author Response · Authors · 2022-08-08
> > > > **Following up**
> > > >
> > > > We are keenly looking forward to your response to our rebuttal. We thought deeply about the concerns raised by you and added several additional sub-sections in our paper discussing the crucial points raised in your review. We also conducted several additional experiments based on your questions and gain valuable insights about our work. We would be happy to answer any additional questions that you might have about our work.
> > > >
> > > > Thank you.

---

### Official Review · Reviewer_V8qo · 2022-07-09

**Rating:** 5
**Confidence:** 3
**Soundness:** 3 good
**Presentation:** 3 good
**Contribution:** 3 good

**Summary:**


This paper studied two types of memorization that impedes the generalization, the heuristic memorization and the example-level memorization, respectively.  This paper proposed that diversity in the activation pattern is linked to indicate these memorizations through information-theoretic measures.

To study the cause of the two types of memorizations, this paper establish the analysis from information theory side, it introduced two information theory measurements: the cross-sample diversity (i.e. entropy) and the cross-neuron diversity (MI). It then showed that heuristic memorization is linked to low cross-sample and cross-neuron diversity, while the example level memorization is linked to the both high value for these two diversities. The authors then demonstrated a potential application of this framework for model selection.


**Questions:**

The applicability of these two metrics across heterogeneous architectures are under-discussed. Based on my understanding, the table 3 in the supple line 565 is only describing architectures within one model setting. If one trained two models with different architectures, could these metrics stil be comparable?

Minors:

line 153:  "This is done" capital T.

**Limitations:**

There is no obvious negative societal impact given the theoretical nature of this work.


**Strengths And Weaknesses:**


This paper is overall well-written in clarity. The attacked problem, model memorization quatification, is a theoretically important task. This paper proposed one way to understand the neural network behavior with grounded tools.


As one limitation, the author claimed in line 264 that this model could be used point-wise to characterize model behavior over individual examples, potentially useful for OOD detection; indeed there is a practical gap before pointwise application, as when people build a single model (drawn one point from the model architecture space) with arbitrary shape, it is easy to measure the entropy and MI but difficult to tell whether these number indicates a high memorization ratio. When people drawn multiple samples from model architecture space, it is also hard to tell whether these meaturements remains descriptive across heterogeneous architectures.

As the metric values are relative references not direct pointer, the authors are encoraged to use a subsection to discuss about the applicabilicable range and the possible issue in practical run.

On the other hand, observed from the results in Fig 2/3, the argued relationship can be outweighted than the variance except when the handcraft ratio alpha is extremely large. Similar observation can be found in Fig 6/7, where the pattenr of the mean of entropy and MI is observable, but the variance remains big. Given this, an independent subsection in the paper for more discussions and adequate claim is recommended.

---

> ### Author Response · Authors · 2022-08-02
> **Response to Reviewer V8qo**
>
> We thank the reviewer for their careful evaluation of our work. We are encouraged to see that the reviewer appreciated the clarity in our presentation and writing. We appreciate that the reviewer acknowledges that our proposed measures are well grounded and provide a new perspective for this important task. Here, we address the reviewer's feedback and questions:
>
> > As one limitation, the author claimed in line 264 that this model could be used point-wise to characterize model behavior over individual examples, potentially useful for OOD detection; indeed there is a practical gap before pointwise application, as when people build a single model (drawn one point from the model architecture space) with arbitrary shape, it is easy to measure the entropy and MI but difficult to tell whether these number indicates a high memorization ratio. …
> As the metric values are relative references not direct pointer, the authors are encoraged to use a subsection to discuss about the applicabilicable range and the possible issue in practical run.
>
>
> The reviewer rightly notes that owing to the relative nature of evaluation with our framework, it could be difficult to place practical applications such as OOD detection. We have now added a new section (appendix E) that discusses possible issues and ideas in a practical run.
>
>
>
> > On the other hand, observed from the results in Fig 2/3, the argued relationship can be outweighted than the variance except when the handcraft ratio alpha is extremely large. Similar observation can be found in Fig 6/7, where the pattenr of the mean of entropy and MI is observable, but the variance remains big. Given this, an independent subsection in the paper for more discussions and adequate claim is recommended.
>
>
> We agree with the reviewer that the large number of values for entropy and MI for each model results in a wide distribution. This could make it challenging for practioners to make comparisons between models that do not differ greatly in their generalization. As per the reviewer's suggestion, we have included a subsection (appendix C.4) that discusses this variance and adjusts the claim accordingly.
>
> > The applicability of these two metrics across heterogeneous architectures are under-discussed [...] If one trained two models with different architectures, could these metrics stil be comparable?
>
> Sensitivity to architectures is an important aspect and we thank the reviewer for bringing this up. Theoretically, we expect that capacity of a network would influence entropy and MI. To test this, we train MLP networks with varying depths for the tasks of Colored MNIST and Shuffled MNIST, and compare the neuron entropy for the obtained networks. The effect of this changing capacity with different memorizing ratios is added and discussed as a new sub-section in appendix C.2 (Figure 9). We observe that our hypothesized trends hold across different models for a given capacity (we’ve validated this for networks ranging from a single to four-layered networks). However, different capacity networks show entropy values in different ranges and hence they are not directly comparable across models of different capacities and can not be used for model selection.

---

> > ### Author Response · Authors · 2022-08-08
> > **Following up**
> >
> > We are keenly looking forward to your response to our rebuttal. We thought deeply about the concerns raised by the reviewer and added several additional sub-sections in our paper discussing the crucial points raised by you. We also conducted additional experiments based on the insightful questions raised in your review. We would be happy to answer any additional questions that you might have about our work.
> >
> > Thank you.

---

### Official Review · Reviewer_hktY · 2022-07-11

**Rating:** 6
**Confidence:** 4
**Soundness:** 3 good
**Presentation:** 4 excellent
**Contribution:** 2 fair

**Summary:**

The paper investigate the detection of memorization. The author categorize the memorization into 2 kinds: heuristic based and example based. The main hypothesis is, compared with the base case, the neural activations of heuristic based memorization has lower entropy and low MI, while the other one has higher entropy and MI. This is empirically and qualitatively validated on both language and vision tasks, and the author provide a quantitative analysis of this metric as a way for model selection.

**Questions:**

1. Is entropy and MI strongly correlated? If so can we just use one of these metric?
2. Can you provide some insights on how this can be useful in designing a regularizer for training?

**Ethics Review Area:**

["I don’t know"]

**Limitations:**

Not that I can see.

**Strengths And Weaknesses:**

Strength:
The paper is easy to follow and well presented. The idea of observing neural activities for memorization is interesting, and I think proposition of MI and inner neurons entropy could be useful for researchers interested this topic. The experiments are solid to show the soundness of the hypothesis.

Weakness:
The main critique for me is that, while the model selection application is given, I am not sure how good that number is. There are some other baselines on ranking models based on generalization metric, e.g., [1], [2]. I would like to see more comparison with some baselines from these work to see the value of model selection.


[1] https://arxiv.org/abs/1912.02178
[2] https://arxiv.org/abs/2012.07976

---

> ### Author Response · Authors · 2022-08-02
> **Response to Reviewer hktY**
>
> We thank the reviewer for their review and are glad to know that they recognize that the aspect of analyzing neural activations using information theoretic approaches is unique to our approach and could be an interesting perspective for this problem.
>
> > I would like to see more comparison with some baselines from these work to see the value of model selection.
>
> We have now performed model selection on several benchmark generalization measures, as suggested by the reviewer. Norm-based measures are popularly used in literature (including the papers referred by the reviewer) as a measure of generalization, and we conduct comparisons with three of them: 2-norm, Frobenuis-norm, and Path-norm. We do this for all our setups across example-level and heuristic memorization (Path-norm is not computed on the transformer models, since no established way exists to do so). On comparing the model selection performance with our information measures, we observe the following:
>
> |                      |                 | Sentiment  Adjectives |    Colored  MNIST   |             | Bias-in-Bios |                 |   Shuffled  MNIST   |    Shuffled  IMDb   |
> |----------------------|-----------------|:---------------------:|:-------------------:|:-----------:|:------------:|:---------------:|:-------------------:|:-------------------:|
> |                      |                 |  Validation Accuracy  | Validation Accuracy | Compression |    TPR-Gap   | Sufficiency-Gap | Validation Accuracy | Validation Accuracy |
> | Information Measures | Mean of Entropy |                  0.80 |                1.00 |        0.47 |         0.20 |            0.20 |                1.00 |                0.60 |
> |                      |    Mean of MI   |                  0.80 |                1.00 |        0.60 |         0.07 |            0.33 |                1.00 |                0.80 |
> |       Baselines      |      2-Norm     |                  0.00 |               -0.20 |        0.33 |         0.60 |            0.87 |                0.80 |                1.00 |
> |                      |  Frobenius-Norm |                 -0.40 |                0.80 |       -0.20 |         0.07 |            0.07 |                1.00 |               -1.00 |
> |                      |    Path-Norm    |                       |                0.80 |             |              |                 |                1.00 |                     |
>
> While these baselines (specifically Frobenius-norm and Path-norm) do  reasonably well in ranking models that perform example-level memorization, they do poorly for heuristic memorization. This follows from the fact that these metrics are conventionally based and tested on the more widely known notion of memorization, that of individual examples. Where 2-norm gives a perfect ranking agreement ($\tau=1$ on Shuffled-IMDb, it is totally uncorrelated ($\tau=0.00$) for Sentiment Adjectives.
> Our information measures hold positive $\tau$ values across all models and tasks, while the baseline measures are inconsistent, varying from extremely low to high values across setups. Frobenius-norm shows perfect negative and positive correlations with Shuffled IMDb and MNIST, respectively, indicating that a high norm value could mean anything depending on the dataset and model.
>
> > Is entropy and MI strongly correlated? If so can we just use one of these metric?
>
> While in our experiments we found these two metrics to be correlated, they theoretically capture different diversity phenomena and therefore we would likely not be able to use these metrics interchangeably.
>
> > Can you provide some insights on how this can be useful in designing a regularizer for training?
>
> Prior work on training dynamics has shown that heuristic memorization and example-level memorization tends to happen at different times during training. Where the former involves the memorization of simple shortcuts that can be derived from the training data, it is more likely to be performed by the model during the early stages of training. At this time, we could enforce a regularizer that minimizes the MI (and maximizes entropy), since the reverse trends are indicative of heuristic learning.  The latter, on the other hand, tends to happen with prolonged training in the later stages, and thus at this stage, the regularizer could be made to maximize MI (and minimize entropy). This pattern could be implemented using a regularizing coefficient that changes its regularizing direction as training progresses. Needless to say, such an exploration would need extensive analysis and experimentation. Where our paper throws light on the potential use of the presented information measures, future work can build upon it and derive more interesting and important use-cases.

---

> > ### Author Response · Authors · 2022-08-08
> > **Following up**
> >
> > We are keenly looking forward to your response to our rebuttal. We conducted experiments based on your requests and they give further insights into our work. We thought deeply about some of the questions you raised and tried to answer them to the best of our ability. We would be happy to answer any additional clarifications that you might have about our work.
> >
> > Thank you.

---

### Official Review · Reviewer_LLo7 · 2022-07-11

**Rating:** 6
**Confidence:** 4
**Soundness:** 3 good
**Presentation:** 3 good
**Contribution:** 2 fair

**Summary:**

This paper proposes to analyse DNNs' generalization behaviours by looking at activation patterns, in particular the entropy of a neuron, and the mutual information between neurons. Experiments show that these measures rank different models similarly to other extrinsic measures (which require labels). This is shown by perturbing labels and training data distributions, leading to two extremes (1) spurious correlations and (2) example memorization.

An interesting finding of this paper is that a "normal" neural network's entropy and MI seems to be in a sweet spot. Too much MI and the network is probably having spurious correlations, too little, and it's probably memorizing individual examples.

**Questions:**

Concerns:

I'm not sure I understand what entropy is being computed, or if it even is an entropy. Shouldn't you want to compute the entropy of the empirical categorical distribution of $A_i$ over the dataset, when $A_i$ is discretized into bins? The sum in the entropy calculation would then be over the number of bins, not the number of examples $S$. $H(A_i)$ should be
$$H(A_i) = \sum_{j=1}^{N_{bins}} p(\hat{a} _ {i}^j) \log p(\hat{a} _ {i}^j)$$
where

$$p(\hat{a}^j_i) = \frac{1}{S}\sum_k^S 1[\mathrm{bin}(a_{(i,k)}) ==  j]$$

Would it be possible to reproduce the plots in the paper with the above computation? -- I'd really like to understand this before accepting this paper.

> [L98] we fit a Gaussian mixture model over all values

Why not just plot the histogram of values? Seems like this could hide some artefacts.

As mentioned above, what models are Table 2's values computed on? Would a practitioner looking to apply this model selection see this by, say, just changing the learning rate or capacity of a model?


**Limitations:**

The authors address social impacts, but limitations are limited to future work rather than fundamental weaknesses of the proposed approach.

**Strengths And Weaknesses:**

Originality: This paper is not the first to analyse entropies and mutual information within neural networks. The specific instantiation and what's being tested are novel, and hopefully informative to the community, but not wholly surprising.

Quality/Clarity: The experiments in the paper are fairly well explained and executed (although I have some concerns), and the thesis is very clear.

Significance: These findings are certainly interesting, but I'm not sure they are groundbreaking either. The authors suggest that the proposed measures can be used to rank models and thus perform model selection, but this result feels a bit forced, since the authors appear to compare models which they specifically train with various algorithmic changes/perturbations that might cause this.

---

> ### Author Response · Authors · 2022-08-02
> **Response to Reviewer LLo7**
>
> We thank the reviewer for their thoughtful feedback. We are glad to see that they appreciated the novelty of our approach, clarity of our thesis and found our experiments well executed. We address the reviewer's comments and questions below:
>
> > the authors appear to compare models which they specifically train with various algorithmic changes/perturbations
>
> We agree that some of our experimental setups are based on synthetically induced artifacts in the training dataset. Such a setup is useful for controlled experiments, as done also in other work on OOD generalization. However, we also experiment with natural setups, namely gender bias in text classification and image classification with contexts (Section 3.2). We observe that our hypothesis holds over these datasets as well. Table 2 further shows how our measures could be a useful tool for model selection—even for instances with naturally occurring memorization.
>
> > Would it be possible to reproduce the plots in the paper with the above computation? -- I'd really like to understand this before accepting this paper.
>
> Thanks for pointing this out. It was an error on our part while writing the equation in the paper. We tried to derive the end formula as an expectation of Shannon's information measure, which led to the summation over samples rather than the number of bins, as it should be. Our results are indeed produced via the correct equation, with a "-1' multiplying factor in the first equation stated by the reviewer, as it should be.
> This can also be verified in the implementation and code that was provided in the supplementarity material with the original submission (appendix I; https://anonymous.4open.science/r/information-reflects-memorization-1). From "utils/information.py" (lines 106--118):
> ```
> def _get_discrete_entropy(X: np.array) -> float:
> 	_, counts = np.unique(X, return_counts=True)
> 	probs = counts / len(X)
> 	[...]
> 	ent = 0.
> 	for i in probs:
> 		ent -= i * np.log(i)
> 	return ent
> ```
> We have updated the equation (Equation 1) and corresponding algorithm (Algorithm 2) in our paper.
>
> > What models are Table 2's values computed on?
>
> We use 3-layered MLP networks for the MNIST tasks, DistilBERT for the tasks on IMDb, and RoBERTa-base on Bias-in-bios. These models are consistent with models from the experiments discussed in the prior sections (Figures 1-7). The Kendall tau correlation coefficient is computed between rankings obtained from different measures (such as the mean of MI or mean of Entropy) with the true rankings of the models (as obtained from the validation performance).
>
> > Would a practitioner looking to apply this model selection see this by, say, just changing the learning rate or capacity of a model?
>
> Sensitivity to hyperparameters and architectures is an important aspect and we thank the reviewer for bringing this up. Theoretically, we expect that capacity of a network would influence entropy and MI. To test this, we train MLP networks with varying depths for the tasks of Colored MNIST and Shuffled MNIST, and compare the neuron entropy for the obtained networks. The effect of this changing capacity with different memorizing ratios is added and discussed as a new sub-section in appendix C.2 (Figure 9). We observe that our hypothesized trends hold across different models for a given capacity (we’ve validated this for networks ranging from a single to four-layered networks). However, different capacity networks show entropy values in different ranges and hence they are not directly comparable across models of different capacities and can not be used for model selection.

---

> > ### Author Response · Authors · 2022-08-08
> > **Following up**
> >
> > We are keenly looking forward to your response to our rebuttal. We believe that we addressed most of the concerns raised by you and conducted additional experiments based on the insightful questions raised in your review. We would be happy to answer any additional clarifications that you might have about our work.
> >
> > Thank you.

---

### Author Response · Authors · 2022-08-02
**General Response**

We thank the reviewers for their thoughtful and detailed feedback. The reviewers appreciated the clarity of our presentation and thesis (R1, R2, R3, R4), the novelty of our approach (R2, R3, R4), and the efficacy and diversity of our experiments (R1, R2, R4).

We would first like to re-emphasize that the main contributions of our work constitutes the analysis of diversity in neural networks representations using methodologies grounded in information theory. Our observation that such measures correlate to the generalization of these networks are unique and can provide a new perspective to generalization. In addition, unlike prior approaches of measuring OOD generalization, our measures work on in-distribution input examples, yet give indications about OOD behaviour. Moreover, while other generalization measures and model selection criteria exist for a single form of memorization alone (usually example-level), our measures are able to provide conclusive rankings for both memorization types covered in the work. Our work also gives way to potential applications such as model regularization and OOD detection based on these information measures. While our work gives way to a large amount of potential applications, introducing the particular methodologies involved in these applications is not the central theme of our work, and rather an exciting future direction.

We recognize that the biggest sticking points were lack of baselines for using our insights for model selection and unclear potential applications of the proposed information measures.

For model selection, we conducted additional experiments to compare with prior generalization measures suggested by R2. We note that our measures outperform these baselines, with a particularly large difference in ranking heuristic memorization models. More details are given in the updated section of our paper (appendix D.2) and in the response to R2 and R4.

Additionally, we envision several potential applications where our insights could be valuable. These applications include: model selection, model improvements through regularizing encoded information as per our insights, OOD detection. We examine one of these applications (i.e., model selection) in detail in this work, and we shed light on the other two applications in response to reviewers’ comments (R3, R4). We also include a new section (appendix E) to discuss ideas and limiting factors for these applications in more detail. Elaborating and exploring the role of these insights in other applications remains a part of the future work.

---

### Meta-Review · Area_Chair_Rbmk · 2022-08-23

**Recommendation:** Accept
**Confidence:** Less certain

**Metareview:**

This paper analyzes the neural network representations using tools from information theory, and show that it could reflect memory patterns and distinguish between "heuristic memorization" and "example-level memorization". The reviewers generally find those findings novel and interesting. There were some concerns such as missing baseline comparison with other model selection criterions. The authors provided additional results in the rebuttals to address those concerns and improved the paper writing according to the reviewers' comments.

**Award:**

No

---

### Decision · Program_Chairs · 2022-09-14

Accept